# The Impact of Foreign Investors on the Stock Price of Korean Enterprises during the Global Financial Crisis

**Yoonmin Kim [†] and Gab-Je Jo *,[†]**

Department of Economics and Finance, Keimyung University, Daegu 42601, Korea; ykim419@kmu.ac.kr
* Correspondence: gabjejo@kmu.ac.kr; Tel.: +82-53-580-5407
† Y.K. is the first author, G.J. is the corresponding author.

**Abstract:** This paper investigates the impact and behavior of foreign equity investment on the price of the nine largest KOSPI (Korea Composite Stock Price Index) enterprises and Samsung Electronics preference stocks in terms of market capitalization during the global financial crisis (2 January 2007 to 30 December 2008). The empirical results indicate that foreign investors show strong, positive feedback trading behavior with regard to the stock price of Samsung Electronics, which is the largest KOSPI enterprise in terms of market capitalization. We also found evidence that the behavior of foreign investors significantly increased volatility in the stock returns of the two largest Korean conglomerates (Samsung Electronics and Hyundai Motors), which account for approximately 25 percent of total KOSPI market capitalization.

**Keywords:** foreign investor groups; individual investor groups; institutional investor groups; volatility; capital market openness; information asymmetry; sustainable capital market; emerging markets

---

## 1. Introduction

There is considerable debate over whether foreign investors stabilize or destabilize domestic stock markets. Foreign investors are often blamed for difficulties in the Korean economy, such as the collapse of both the Won and the stock market. A check was conducted to ascertain whether foreign investors increased the volatility of daily stock returns more than domestic investors in the nine largest KOSPI (Korea Composite Stock Price Index) enterprises and Samsung Electronics preference stocks (Samsung Electronics, Samsung Electronics (Preference Share), Hyundai Motors, KEPCO (Korea Electric Power Corporation), POSCO (Pohang Iron and Steel Company), SK Hynix, NAVER, Amore Pacific, and Samsung C&T (Construction & Trading Corporation)) during the global financial crisis.

Most research regarding the behavior of each investor group in the Korean stock market took place during the Asian financial crisis period, and not particularly the global financial crisis. Inclusive, the research was performed at the state or industrial level. Therefore, we decided to investigate behavior at the firm level (the price of the nine largest KOSPI enterprises and Samsung Electronics preference stocks) during the recent global financial crisis period from 2 January 2007 through 30 December 2008.

This research is a comparative analysis of the different roles and impact of foreign equity investments on market volatility in emerging Asian markets. Our research presents advice for maintaining sustainable capital market development in emerging markets. In our paper, "sustainable capital market development" means stable capital market development that prevents possible speculative attacks and sudden breaks in capital inflow. Our research could serve as a wakeup call to prevent the outflow of national wealth in emerging markets. Therefore, we maintain that our research can contribute to sustainability in emerging Asian markets. This paper suggests that

researchers, policy officials, and market participants should find it useful to keep both approaches in their tool kits for analysis.

In order to study the interrelationship between variables, the empirical procedure began with the study of dynamic relationships between interested variables. In order to study the behavior of equity flows and their effects on the stock prices of the nine largest Korean enterprises and Samsung Electronics preference shares, vector autoregression (VARS) was employed to investigate the dynamic relationship between daily percentage changes in the volatility of the stock price and the daily percent changes in the net buy ratio (NBR) for Korean institutions, Korean individuals, and foreign investors. Second, the Granger-causality was applied to explore the casual relationship in each of the variable systems. Lastly, the plotted impulse response function of the variable system was also employed.

The paper proceeds as follows: In Section 2 we describe the research background and our motivation. In Section 3 the literature is reviewed to note any destabilization in the domestic stock market caused by foreign investors. In Section 4 we investigate whether foreign investors engage in positive feedback trading and destabilization of the Korean stock market at the firm level and not at the state or industry level. We explain our findings in Section 5.

## 2. Research Background

Korea is considered an emerging market that has experienced large foreign capital outflows twice, during both the Asian crisis in 1997 and the global financial crisis in 2008. Nonetheless, the Korean economy stands as the 12th largest economy in the world based on GDP in 2017. The Korean stock market started the liberalization process in early 1992, lifting most foreign ownership restrictions in May 1998 due to the IMF bailout. Since then, the Korean economy successfully overcame the 1997 Asian crisis and restructured the country's economy. In this regard, the Korean market provides a good case for evaluation of the impacts and roles of foreign investors in emerging markets.

In 2004, the proportion of Korean market capitalization held by foreign investors hit the maximum, which was 40.5 percent with the proportion decreasing. In 2018, the foreign equity ownership was about 31.3 percent of KOSPI total market capitalization. However, as shown in Figure 1, the share of foreign investors in Korean blue chips was generally higher than the regular KOSPI stocks (except KEPCO and SK Hynix). Moreover, there was a continuous increase in proportion. Therefore, we decided to perform the research at the firm level and not at the state or industrial level.

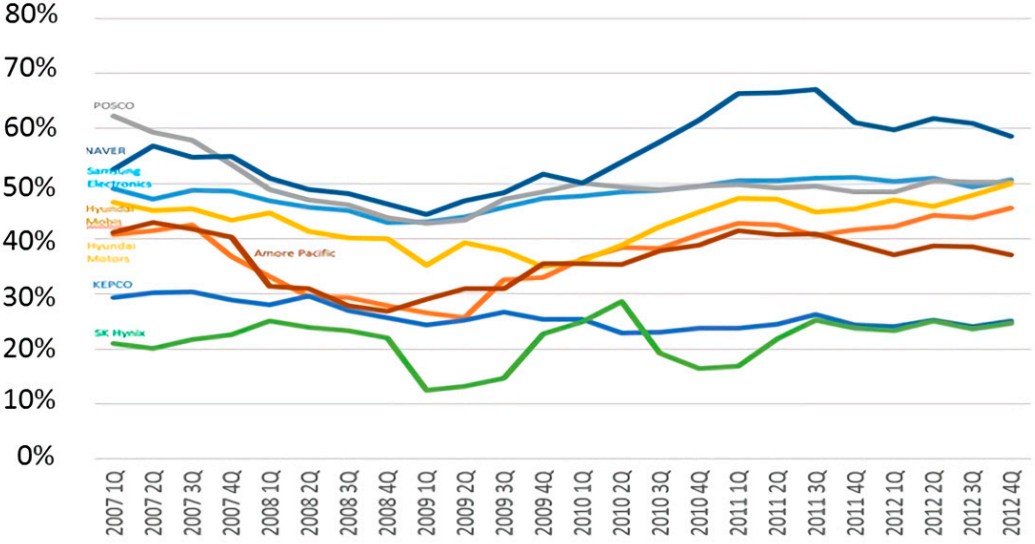

**Figure 1.** Foreigner's share of Korean Enterprises' stock.

## 3. Literature Review

Choe et al. [1] examined the impact foreign investors had on stock returns in Korea from 30 November 1996 to the end of 1997, using order and trade data. The authors classified buying and selling investors into three categories: Korean individual investors, Korean institutional investors, and foreign investors based on data availability. The authors measured abnormal returns for 11 five-minute intervals that were centered on the intervals with large foreign trades of the stock in their sample. For the second test, days were used instead of five-minute intervals. The authors followed the approach of Lakonishok et al. [2] and Wermers [3] to estimate positive feedback trading and herding. This approach was used to compute the herding measure, by using daily horizon and treating each trade on a given day by different investment groups. (The herding measure is computed as $|p_{it} - \mathrm{E}(p_{it})| - \mathrm{E}|p_{it} - \mathrm{E}(p_{it})|$, where $p_{it}$ is the proportion of foreign investors buying stock $i$ on day $t$ among all foreign investors trading that stock on that day and $\mathrm{E}(p_{it})$ is the expected proportion of foreign investors buying on day $t$ relative to all foreign investors. $\mathrm{E}|p_{it} - \mathrm{E}(p_{it})|$ is an adjustment factor computed assuming that in the absence of herding, the number of foreign investors with net purchases follows a binomial distribution.) The authors documented strong evidence of positive feedback trading and herding by foreign investors in South Korea before the Asian financial crisis period; however, these patterns disappeared during the crisis. The authors of References [2,3] concluded that foreigners did not destabilize the Korean stock market during the Asian crisis.

Nimitvanich [4] examined the daily data from the SET index (the composite index for the Thai stock exchange) and net foreign flows from 1995 to 2010. Nimitvanich applied vector autoregression (VARS) to investigate the relationship between the daily return of the SET index and the net foreign flow. He found strong evidence that foreign investors follow positive feedback or momentum trading because foreign flows are positively related to prior returns. Moreover, he found that foreign investors had some ability to forecast local equity returns.

It has been suggested that the causes of Korean stock market volatility were double-dip worries, the fear factor, foreign economy uncertainty, and lack of political leadership. However, foreign investors are still harshly blamed for the difficulties in the Korean economy such as the collapse of the Won and the stock market. High volatility and reversibility caused by foreign investors are very important issues for the Korean stock market. Jo [5] found evidence that equity investment activities by foreigners were more reversible than domestic investment during the Asian crisis. (Using two stage least squares (TSLS), Jo [5] investigated the impact of the absolute value of net purchases by three investor groups on daily stock market volatility, measured using the generalized autoregressive conditional heteroskedasticity (GARCH) model.) In addition, he found that foreign equity investors tend to cause higher volatility in the market than domestic investors. Furthermore, Jo [5] found that foreign investors led the withdrawal from the market (net sellers), before the IMF financing program. However, after the IMF program, the situation reversed itself. Foreigners became primary net buyers. According to Hamao and Mei [6], there is no evidence that equity investment by foreigners increased the Japanese stock market volatility more than domestic investment during the Asian financial crisis. However, Jo [5] argued that as an emerging market, the Korean market was much more vulnerable to sudden capital flight than the firmly established Japanese market, one that avoided crisis.

Choe et al. [7] contend that domestic investors have a strong home bias. Therefore, domestic investors overweight the domestic market in their portfolios. However, foreign investors usually have international expertise and talent, as well as considerable local resources. Moreover, Kho [8] argued that domestic investors did not have the advantage foreign investors had in KOSPI. Foreign investors performed better because they had international expertise with information sources and credible local information sources with no home bias, permitting them access to macroeconomic information for index options. However, their advantages were due to more advanced investment techniques and better corporate governance systems in their own countries. Moreover, foreign investors had much more difficulty accessing firm-specific information for KOSPI because of their physical and linguistic barriers.

Joe and Oh [9] investigated the behavior of foreign investors in the Korean stock market after the 1997 Asian financial crisis (1999–2014). The authors analyzed the industrial distribution of foreign ownership. Joe and Oh [9] insisted that foreign investors showed a preference for large, profitable, highly liquid, and growth firms, as well as those with large boards. However, Chaebol (Korean conglomerates) firms were not attractive to foreign investors. In other words, foreign investors were not blindly purchasing Chaebol stocks (blue chips in KOSPI), and they considered many financial factors in stock purchase. In the end, the authors of Reference [9] insisted that foreign investors achieved success as financial investors in Korea.

We also reviewed the literature regarding information asymmetry in the Korean stock market. We classified the literature into three categories: foreign investor information superiority, partial superiority and partial inferiority, and inferiority of foreign investor information relative to domestic investors.

### 3.1. Foreign Investor Information Superiority

Ahn, Kang, and Ryu [10], Eom, Hahn, and Sohn [11], Hong and Shin [12], Ko and Kim [13], Ko and Lee [14], and Oh and Hahn [15] argued that foreign investors perform better than domestic investors due to their better expertise and talent. In other words, the superior performance of foreign investors primarily comes from their informational advantage in their asset allocation strategies and their tendency to buy prior to positive and sell prior to negative earnings surprises, while domestic investors do the opposite. Furthermore, according to Choe, Kho, and Stulz [7], if foreigners are more sophisticated, they might perform better in countries with open stock markets (large shares of foreign investors) but not in markets where native only trading is more prevalent.

### 3.2. Partial Superiority and Partial Inferiority

Kang and Stulz [16] and Oh and Hahn [17].

Domestic investors are winners of intraday trading due to a short-lived informational advantage; however, global brokerages (foreign investors) are better at long-term position trading. Therefore, the combination of local information and global expertise can lead to higher profit.

### 3.3. Foreigner Investor Information Inferiority

Choe, Chung, and Lee (2008) [18], Choe, Kho, and Stulz [19], Kang, Lee, and Park [20], Kho and Kim [21], and Park, Bae, and Cho [22].

According to the articles reviewed, foreign money managers often buy at higher prices than domestic investors and sell at lower prices for medium and large trades due to foreign investor return-chasing behaviors in the Korean stock market. Foreign investors in KOSPI perform worse than domestic institutions and individuals because they pay the least amount of attention to temporary component-driven price changes (investors whose expectation changes serially correlates response to price change). All of the literature presented thus far mentions that foreign investors have longer investment horizons than domestic investors.

## 4. Empirical Results

The buy and sell amount was collected from Samsung's fnguide.com for foreign, institutional, and individual investor groups, along with the stock prices of the nine largest KOSPI enterprises and Samsung Electronics preference stocks by market capitalization. The research dated from 2 January 2007 through 30 December 2008. The first date was chosen based on the sudden increase in spillover impact from the economies of the Southern European countries of Portugal, Italy, Greece, and Spain (PIGS) to the Korean financial market. While evidence of the subprime mortgage crisis in the United States became public at that time, the magnitude of the problem was not appreciated until after the failure of Lehman Brothers, when expectations emerged that the crisis would spread to the emerging

market countries. Koreans sank into serious financial turmoil after the Lehman Bankruptcy until the end of 2008, so we set 30 December 2008 as the research end date.

We used NBR (net buy ratio) to measure the investment patterns of different investor groups in the Korean Stock Exchange during the global financial crisis. The NBR for an investment group is calculated by subtracting the sell amount from the buy amount and dividing by the total trade amount, which is the sum of the buy and sell amounts. (In addition, see Song, Yang, & Oh [23].) The NBR is defined as:

$$NBR_{t,i} = \frac{(Buy\ amount)t,i - (Sell\ amount)t,i}{(Buy\ amount)t,i + (Sell\ amount)t,i} \tag{1}$$

where the NBR is that of group *i* on day *t*.

Grinblatt and Keloharju [24] and Griffin et al. [25] argued that the NBR could capture both directions (buying and selling) of investor trading patterns and their relative magnitudes. Therefore, the NBR is appropriate for explaining the inclination of investors to buy and sell, rather than using the net buy amounts or total trading amounts. (NBR > 0: the stock buying of an investor group. NBR < 0: the stock selling of an investor group).

As we can see in the Figure 2, the Korean stock market experienced massive capital outflows during the global financial crisis in 2008, which led to a severe crunch until early 2009. Unlike in 1997, the Korean capital market remained in good condition with sound corporate performance and ample foreign currency reserves. However, hot money flowing into Korean bond and stock markets had already reached 40 trillion won by the end of 2010. (See Financial Supervisory Service of Korea, Reference [26].)

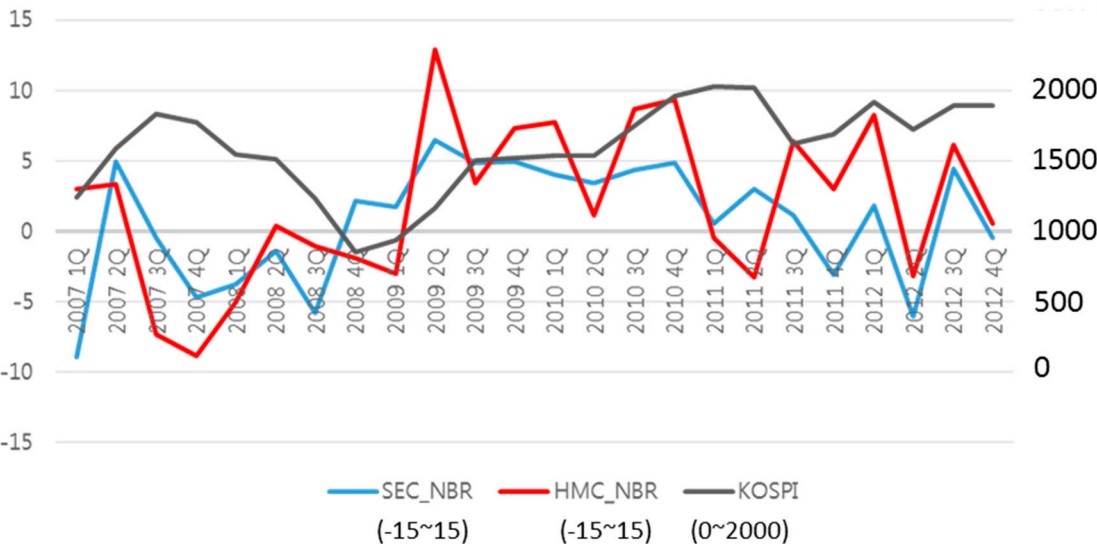

**Figure 2.** KOSPI index and NBR (Net Buy Ratio) for Samsung Electronics and Hyundai Motors.

Next, we used conditional variance, employing Bollerslev's [27] GARCH (generalized autoregressive conditional heteroskedasticity) model to measure the volatility of each stock's daily returns. The reason the GARCH model was employed in this research is due to its effective estimation, alleviating problems of changing variance (heteroscedasticity) of a high frequency time series with autocorrelation, particularly in finance. Also, GARCH model's parsimonious representation was enough to consider found ARCH effects.

Conditional variance was derived from the weighted average of lagged squared residuals at time *t* from an appropriate model of each stock's daily return.

$$\gamma_t = u + \sum_{i=1}^{p} \phi_i \gamma_{t-i} + \sum_{i=1}^{p} \delta_i \varepsilon_{t-i} + \varepsilon_t$$
$$\varepsilon_t | (\varepsilon_{t-1}, \varepsilon_{t-2}, \ldots\ldots\ldots) \sim N(0, \sigma_t^2)$$
$$\sigma_t^2 = \alpha_0 + \alpha_1 \varepsilon_{t-1}^2 + \alpha_2 \sigma_{t-1}^2$$

(2)

where $\gamma_t$ is each stock's return represented as log first difference of each stock's daily price, *u* is a drift term, and $\varepsilon_t$ is the white-noise process. $\phi_i$ and $\delta_i$ are coefficients. ARMA (1,1) was chosen as the model that best fits each stock's returns. $\sigma_t^2$ is a conditional variance, which is a function of $\varepsilon_{t-1}^2$ and $\sigma_{t-1}^2$. $\alpha_0$ is an intercept term, $\alpha_1$ and $\alpha_2$ are coefficients. Thus, the conditional variance for stock market volatility was derived from estimated residuals by estimating the two equations above simultaneously; the conditional mean equation and the conditional variance equation.

For an empirical model, we used the impulse response function (IRF) in a vector autoregression model (VAR). The VAR model can be expressed as follows:

$$B_0 X_t = \beta + \sum_{i=1}^{3} B_i X_{t-i} + \varepsilon_t$$

(3)

where lags were selected by the Akaie Information Criteria (AIC) and the vector *X* included the tree variables. We used NBR for the investment group volatility of each stock's daily returns ($Vol_t$), and each stock's daily return ($Return_t$) ($NBR_t$) ($NBR_t\_For$, $NBR_t\_Inst$, and $NBR_t\_Ind$ indicates $NBR_t$ for foreign, institutional, and individual investment groups, respectively). $B_0$ and $B_i$ are matrices of coefficients, where $\beta$ denotes intercept terms, and $\varepsilon$ denotes the vector of serially and mutually uncorrelated structural innovations.

In order to investigate the effect of the NBR for an investment group on the volatility of each stock's daily returns, the impulse response function (IRF) was employed in this analysis. We carried out the empirical analysis for the global financial crisis period from 2 January 2007 to 30 December 2008.

The IRF estimates the responses for current and future endogenous variables of a one-time shock on the variables in the VAR system. The IRF can be technically described in vector MA($\infty$) form as follows:

$$X_t = \mu + \varepsilon_t + \Psi_1 \varepsilon_{t-1} + \Psi_2 \varepsilon_{t-2} + \Psi_3 \varepsilon_{t-3} \ldots$$

(4)

where $X_t$ is a vector containing the endogenous variables and $\mu$ is the mean of $X_t$. The matrix $\Psi_s$ can be expressed as $\partial X_{t+s} / \partial \varepsilon'_t = \Psi_s$. The row *i* and column *j* element of $\Psi_s$ indicate the impact of a one-unit increase in the jth variable's innovation at date $t(\varepsilon_{j,t})$ on the *i*th variable at time $t + s(X_{i,t+s})$. The coefficients sets $\partial X_{i,t+s} / \partial \varepsilon'_{j,t}$, are the IRFs that show the response of $X_{i,t+s}$ to a one-time impulse in $X_{j,t}$ when all other variables are constant.

As shown in Table 1, according to the unit root test these variables are found to be stationary. Thus, we do not need to specify the first difference of the logarithm.

**Table 1.** Unit Root Tests [++].

| Variables | Augmented Dickey-Fuller Test Statistic | |
| :---: | :---: | :---: |
| | **Samsung Electronics** | **Hyundai Motors** |
| $Vol_t$ | −4.05 *** | −3.42 ** |
| $Return_t$ | −17.17 *** | −22.88 *** |
| $NBR_t\_For$ | −10.30 *** | −11.75 *** |
| $NBR_t\_Inst$ | −8.88 *** | −13.44 *** |
| $NBR_t\_Ind$ | −13.95 *** | −12.82 *** |

[++] ** and *** indicate statistical significance at 5 percent, and 1 percent, respectively. Trend and intercept are not included in the ADF equation. The ADF test is applied to the period from 2 January 2007 to 30 December 2008.

According to the results of this analysis, the case of Samsung Electronics and Hyundai Motor revealed statistically significant results even though seven of the other enterprises and Samsung Electronics preference stocks did not have statistically significant results. While these are just two companies, these results have highly significant meaning because the two enterprises form approximately 25 percent of total KOSPI market capitalization (Samsung Electronics: 18.86% and Hyundai Motors: 5.23%) as we can see in Figure 3.

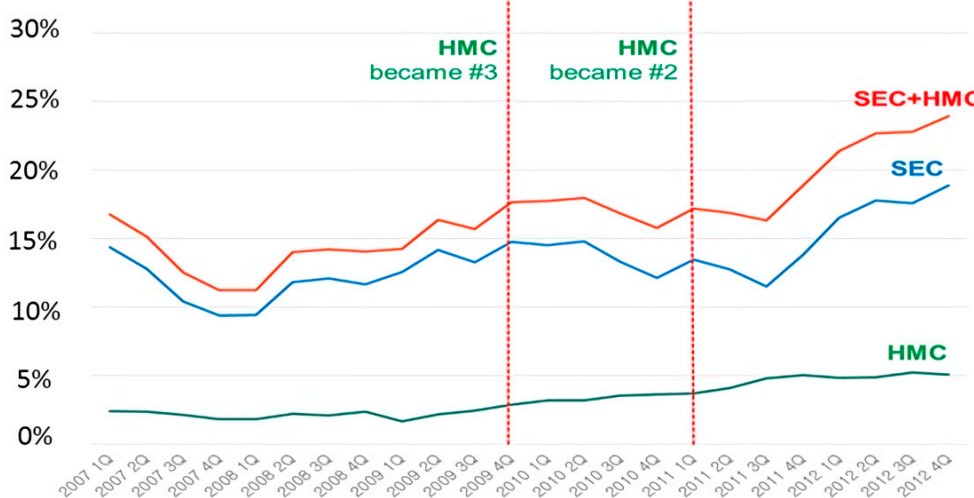

**Figure 3.** Relative Importance of Samsung Electronics and Hyundai Motors in KOSPI.

Figures 4–9 show the results of the IRF. The variable NBR_SAM_FOR is NBR for foreign investors to SEC (Samsung Electronics), RETURN_SAM is Samsung stock return, VOL_SAM is Samsung stock return's volatility, NBR_SAM_INST is NBR for institutional investors to SEC, NBR_SAM_IND is NBR for individual investors to SEC, NBR_HCAR_FOR is NBR for foreign investors to HMC (Hyundai Motors), and NBR_HCAR_INST is NBR for institutional investors to HMC. As indicated, in the Figures, as the "response to Cholesky One S.D. Innovations ±2 S.E.", the IRF results show the response of $X_{i,t+s}$ to a one standard deviation impulse of error term in $X_{j,t}$ of Equation (3), with two standard deviation confidence interval (±2 S.E.). The impulse response period reflects short-term effects from one day to 10 days, which corresponds to this paper's objectives because financial markets rolled heavily in the short-term during the financial crisis. As shown in Figure 1, when the impulse is NBR for foreign investors to SEC, the response of the SEC stock return's volatility is significantly positive until the third period. This is evidence that the SEC stock market volatility was affected by foreign equity investment. That is, foreign investment significantly increased the SEC stock return's volatility.

Figures 4–6 display the impulse response of Samsung Electronics (SEC) stock return's volatility to the SEC stock return, NBR for foreign investors to SEC, NBR for institutional investors to SEC, and NBR for individual investors to SEC during the sample period 2 January 2007 to 30 December 2008.

As shown in Figure 4, when the impulse is NBR for foreign investors to SEC, the response of the SEC stock return's volatility was significantly positive up to the third period. This is evidence that the SEC stock market volatility was affected by foreign equity investment. That is, foreign investment significantly increased the SEC stock return's volatility.

Figure 4 also shows that when the impulse was the SEC stock return, the response of the NBR for foreign investors to SEC was significantly positive up to the third period. However, as shown in Figures 5 and 6, when the impulse was the SEC stock return, the response of NBR for domestic investors to SEC was significantly negative. This is evidence of feedback trading behavior by foreign investors to SEC stock.

Furthermore, Figure 6 indicates that when the impulse is the SEC stock return, the SEC stock return's volatility shows a significant response, with a negative sign. This result means there is volatility asymmetry. That is, the SEC stock return's volatility is higher when the SEC stock return declines.

Figures 7–9 show the impulse response of HMC stock return's volatility to HMC stock return, NBR for foreign investors to HMC, NBR for institutional investors to HMC, and NBR for individual investors to HMC during the global financial crisis period.

According to Figure 7, when the impulse is NBR for foreign investors to HMC, the response of HMC stock return's volatility is significantly positive, up to the sixth period. This is evidence that foreign equity investment significantly increased HMC stock return's volatility.

In addition, Figures 7–9 indicates that when the impulse is HMC stock return, HMC's stock return volatility shows a significantly negative response, which is evidence of volatility asymmetry. That is, HMC stock return's volatility is higher when the HMC stock return declines.

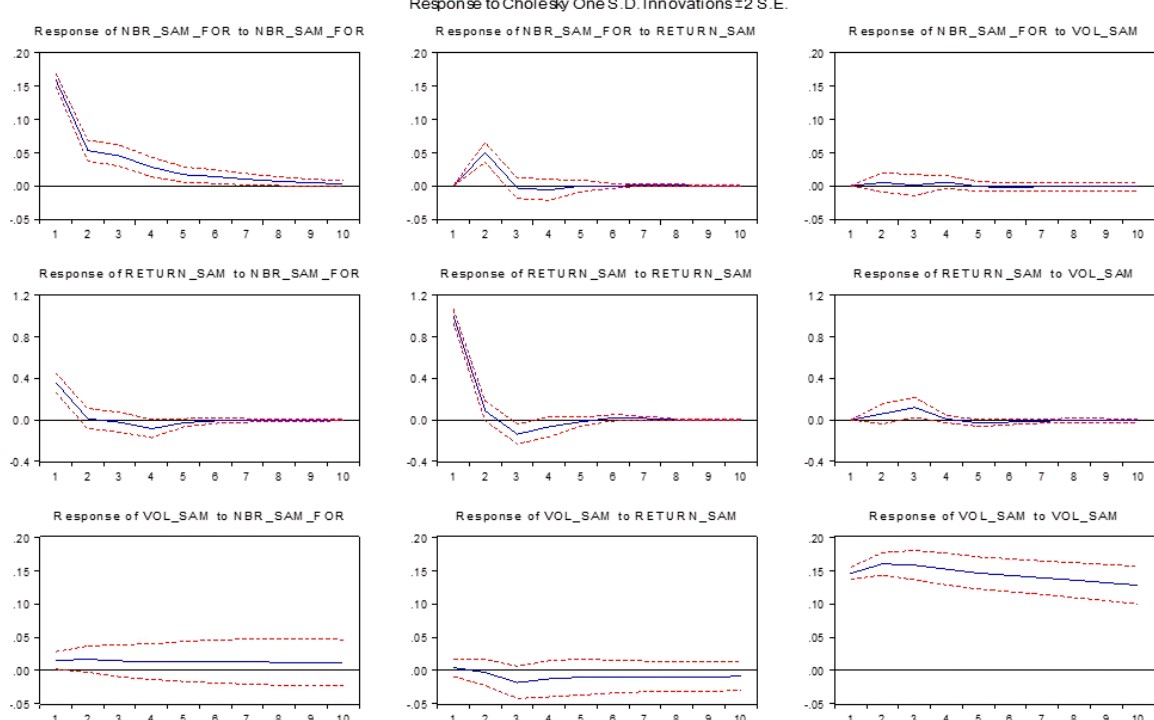

**Figure 4.** IRF results for foreign investment groups to Samsung Electronics.

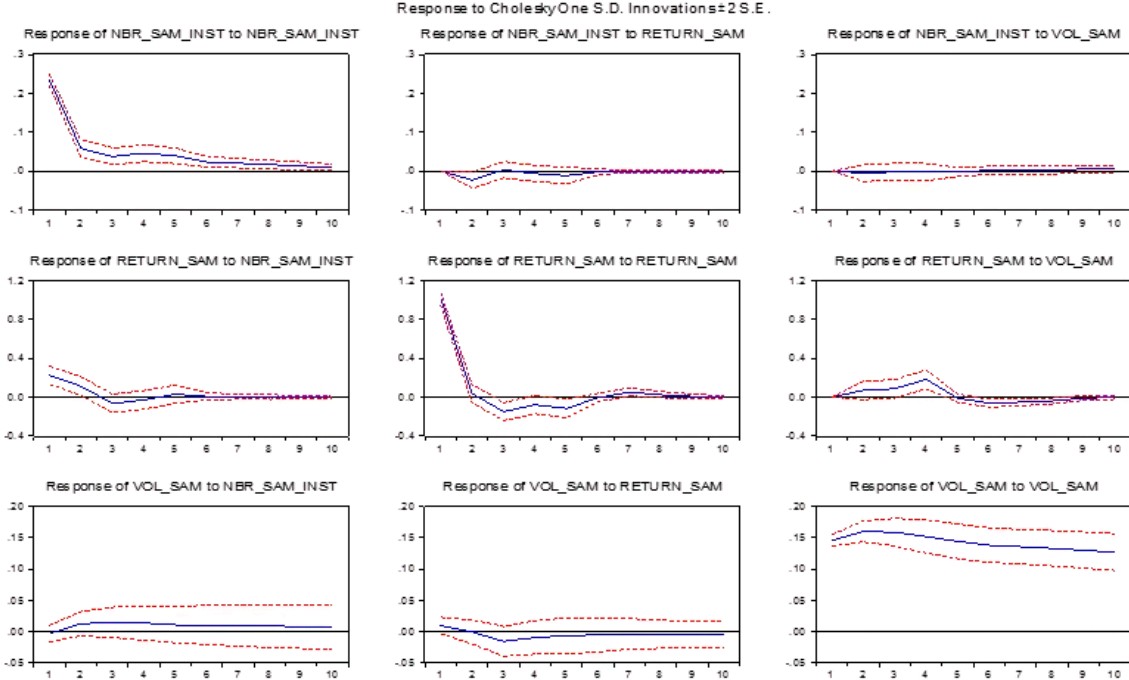

**Figure 5.** IRF results for institutional investment groups to Samsung Electronics.

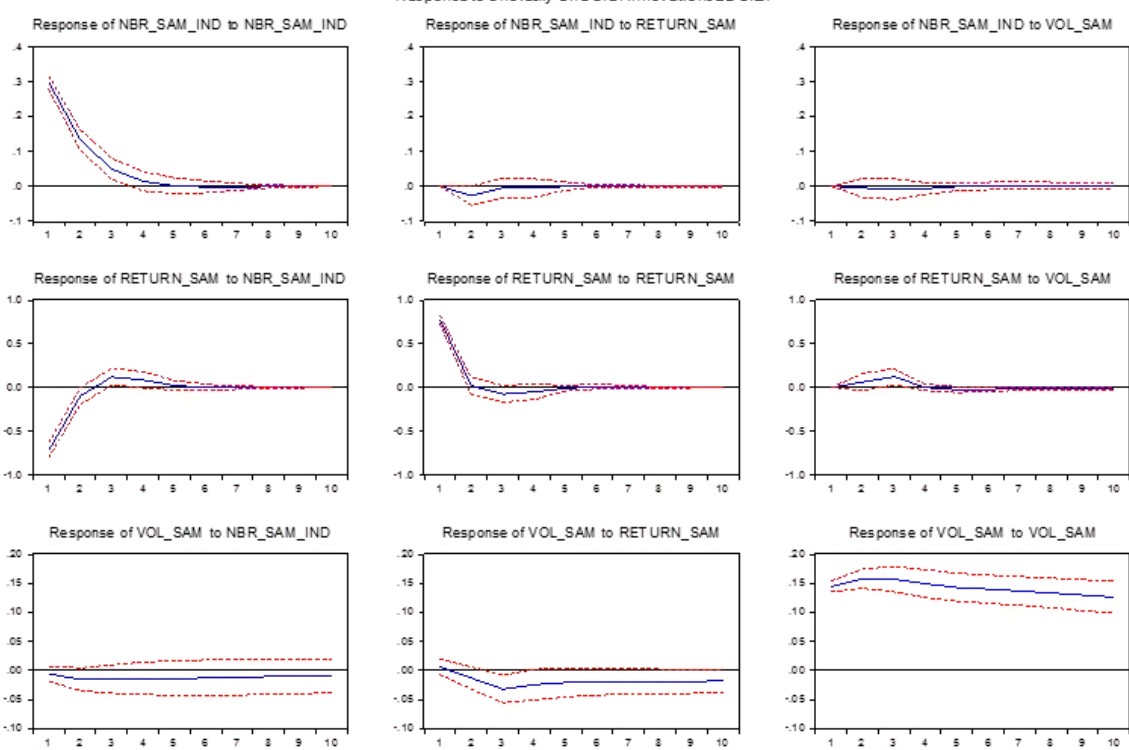

**Figure 6.** IRF results for individual investment groups to Samsung Electronics.

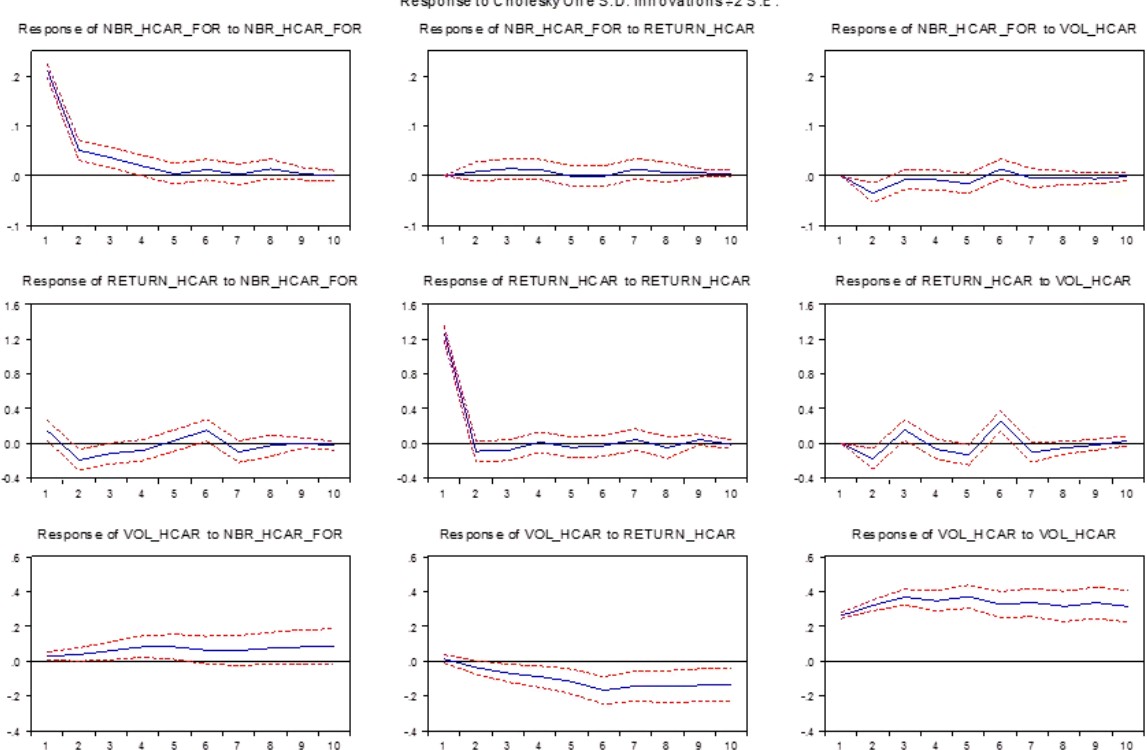

**Figure 7.** IRF results for foreign investment groups to Hyundai Motors.

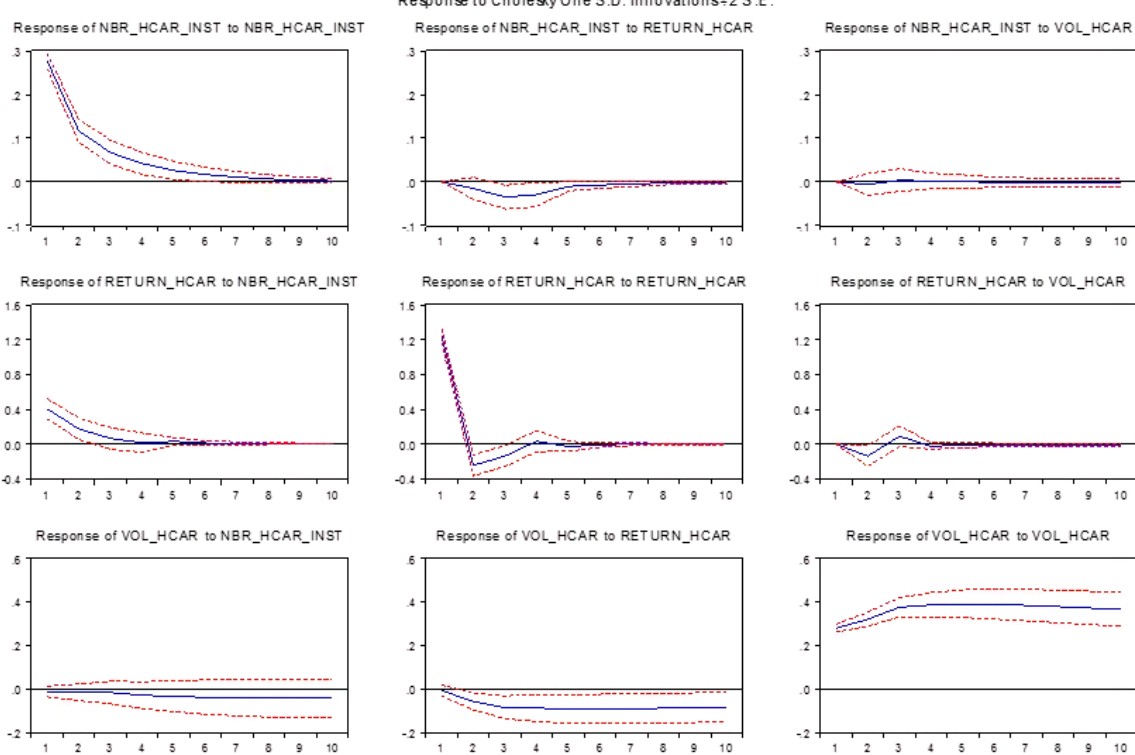

**Figure 8.** IRF results for institutional investment groups to Hyundai Motors.

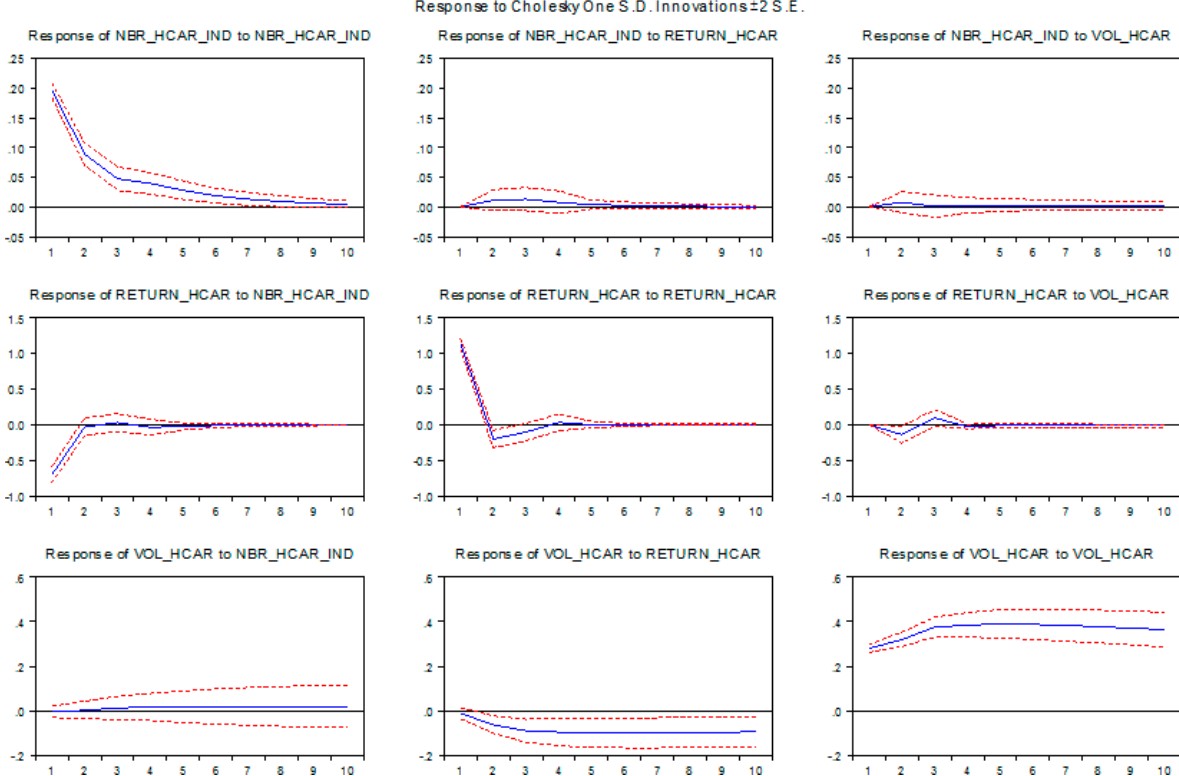

**Figure 9.** IRF results for individual investment groups to Hyundai Motors.

## 5. Conclusions

We reviewed major studies in finance literature accumulated over the last two decades regarding the impact and roles of foreign portfolio investors in Korea. This research will contribute to the literature by organizing previous studies in Korea, making them verifiable and comparable to each other, and eventually narrowing the gap between the research performed at the firm level and research performed primarily at the state level in Korea.

This paper examined the effects of equity investment by foreign investors on the volatility of the nine largest KOSPI market capitalization enterprises' stock price during the global financial crisis. The results indicate that it is meaningful to distinguish between portfolio investment by foreign investors and investment by residents.

In Korea, individual investors are jokingly called "ant warriors" because they face a slim chance of hitting a jackpot due to their lack of information and ability to analyze it. The investments of individual investors were short term and they tended to sell stocks right after making a small profit, while the institutional and foreign investors picked up blue-chip stocks sold by individuals. When the stock market recovered, individual investors who sold their stocks returned, buying shares at relatively high prices. The return of individuals raised stock prices further, causing institutional and foreign investors to sell their shares. Then, the stock prices plunged again, seriously affecting individual investors. According to the statistical results presented in this paper, the SEC stock return's volatility was higher when the SEC stock returns of individual investment groups declined.

In the Korean stock market, one of the most important roles of institutional investors is to prevent capital flight or sudden stops caused by speculative attacks by foreign hedge funds during a financial crisis. In this paper, we show that individual investors bought stocks fervently when the volatility of SEC stock return increased. We present strong evidence of positive feedback trading by foreign investors and negative feedback trading by domestic investors (institutional and individual investment groups) in the case of SEC stocks during the global financial crisis.

Overall, we found some evidence that during the global financial crisis, foreign equity investment significantly affected the stock return's volatility of South Korea's two main companies, Samsung Electronics and Hyundai Motors. We also found that the stock return's volatility concerning these two main companies was higher when the stock return declined. Consequently, we found that equity investment by foreigners in the Korean stock market tended to increase market volatility levels more than investment by residents during the research term.

**Author Contributions:** Conceptualization, Y.K.; methodology, G.-J.J.; software, G.-J.J.; validation, Y.K. and G.-J.J.; formal analysis, G.-J.J.; investigation, Y.K.; resources, Y.K.; data curation, Y.K.; writing—original draft preparation, Y.K.; writing—review and editing, Y.K. and G.-J.J.; visualization, Y.K. and G.-J.J.; supervision, Y.K. and G.-J.J.; project administration, Y.K.; funding acquisition, Y.K.

**Funding:** This research was funded by Keimyung University, grant number 20160490" and "The APC was funded by Keimyung University".

**Acknowledgments:** This research was supported by the Keimyung University Research Grant of 2016.

**Conflicts of Interest:** The authors declare no conflict of interest. The funders had no role in the design of the study; in the collection, analyses, or interpretation of data; in the writing of the manuscript, or in the decision to publish the results.

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
