# Peer review of "The Impact of Foreign Investors on the Stock Price of Korean Enterprises during the Global Financial Crisis"

_sustainability, doi:10.3390/su11061576_

Round 1

Reviewer 1 Report

Please find my comments in the attachment.

Author Response

Dear, reviewer1

Thank you for inviting us to submit a revised draft of our manuscript entitled, “The Impact of Foreign Investors on the Stock Price of Korean Enterprises during the Global Financial Crisis” to Sustainability. We also appreciate the time and effort you and each of the reviewers have dedicated to providing insightful feedback on ways to strengthen our paper. Thus, it is with great pleasure that we resubmit our article for further consideration. We have incorporated changes that reflect the detailed suggestions you have graciously provided. We also hope that our edits and the responses we provide below satisfactorily address all the issues and concerns you and the reviewers have noted.

To facilitate your review of our revisions, the following is a point-by-point response to the questions and comments delivered in your letter.

Reviewer 2 Report

This paper investigates the impact of foreign investors on the stock price of Korean enterprises during the global financial crisis, although the writing is fine, the paper needs to significantly address the issues before being considered for potential publication

 there is no discussion regarding the background information about the Korean stock market

the introduction needs to be significantly improved by talking about the significance, importance and contribution of the current study.

there is no methodology and data section/needs to restructure the results section.

the results section needs to discuss the finding in a more critical way by trying to link the current study with empirical literature.

in the conclusion, there is no discussion regarding the limitations of the current study and areas of future research. 

Author Response

Dear, reviewer2

Thank you for inviting us to submit a revised draft of our manuscript entitled, “The Impact of Foreign Investors on the Stock Price of Korean Enterprises during the Global Financial Crisis” to Sustainability. We also appreciate the time and effort you and each of the reviewers have dedicated to providing insightful feedback on ways to strengthen our paper. Thus, it is with great pleasure that we resubmit our article for further consideration. We have incorporated changes that reflect the detailed suggestions you have graciously provided. We also hope that our edits and the responses we provide below satisfactorily address all the issues and concerns you and the reviewers have noted.

To facilitate your review of our revisions, the following is a point-by-point response to the questions and comments delivered in your letter.

Round 2

Reviewer 1 Report

Please find my review in the attachment.

Author Response

Dear, reviewer1

Thank you again for inviting us to submit a revised draft of our manuscript entitled, “The Impact of Foreign Investors on the Stock Price of Korean Enterprises during the Global Financial Crisis” to your publication, Sustainability. Again we appreciate the time and effort your publication staff and each of the reviewers dedicated to provide us with insightful feedback on ways to strengthen our paper. Thus, it is with great pleasure that we resubmit our article for further consideration. We have incorporated changes that reflect the detailed suggestions you have graciously provided. We also hope that our edits and the responses provided below satisfactorily address all of the issues and concerns you and the reviewers have noted.

To facilitate your review of our revisions, the following is a point-by-point response to the questions and comments delivered in your letter.

Reviewer 2 Report

I am fine with the revision.

Author Response

Dear, reviewer2

Thank you again for inviting us to submit a revised draft of our manuscript entitled, “The Impact of Foreign Investors on the Stock Price of Korean Enterprises during the Global Financial Crisis” to your publication, Sustainability. Again we appreciate the time and effort your publication staff and each of the reviewers dedicated to provide us with insightful feedback on ways to strengthen our paper.

This manuscript is a resubmission of an earlier submission. The following is a list of the peer review reports and author responses from that submission.